# Effect of *Borrelia burgdorferi* on the Expression of miRNAs in Breast Cancer and Normal Mammary Epithelial Cells

**DOI:** 10.3390/microorganisms11061475

**Published:** 2023-06-01

**Authors:** Ananya Debbarma, Miranda Mansolf, Vishwa A. Khatri, Justine A. Valentino, Eva Sapi

**Affiliations:** Lyme Disease Research Group, Department of Biology and Environmental Science, University of New Haven, 300 Boston Post Road, West Haven, CT 06516, USA; adebb1@unh.newhaven.edu (A.D.); mmans3@unh.newhaven.edu (M.M.); vkhat1@unh.newhaven.edu (V.A.K.); jvale7@unh.newhaven.edu (J.A.V.)

**Keywords:** breast cancer, *Borrelia burgdorferi*, infection, microRNA, biomarkers

## Abstract

Breast cancer is one of the leading causes of death in women worldwide. Recent studies have demonstrated that inflammation due to infections with microorganisms could play a role in breast cancer development. One of the known human pathogens, *Borrelia burgdorferi*, the causative agent of Lyme disease, has been shown to be present in various types of breast cancer and is associated with poor prognosis. We reported that *B. burgdorferi* can invade breast cancer cells and affect their tumorigenic phenotype. To better understand the genome-wide genetic changes caused by *B. burgdorferi*, we evaluated the microRNA (miRNA or miR) expression profiles of two triple-negative breast cancer cell lines and one non-tumorigenic mammary cell line before and after *B. burgdorferi* infection. Using a cancer-specific miRNA panel, four miRNAs (miR-206, 214-3p, 16-5p, and 20b-5p) were identified as potential markers for Borrelia-induced changes, and the results were confirmed by quantitative real-time reverse transcription (qRT-PCR). Among those miRNAs, miR-206 and 214 were the most significantly upregulated miRNAs. The cellular impact of miR-206 and 214 was evaluated using DIANA software to identify related molecular pathways and genes. Analyses showed that the cell cycle, checkpoints, DNA damage–repair, proto-oncogenes, and cancer-related signaling pathways are mostly affected by *B. burgdorferi* infection. Based on this information, we have identified potential miRNAs which could be further evaluated as biomarkers for tumorigenesis caused by pathogens in breast cancer cells.

## 1. Introduction

Breast cancer is one of the leading causes of death in women, exhibiting a severe prognosis and high mortality rate [1]. Advancements in the stages of breast cancer make the prevention of the disease more complex; hence, detecting breast cancer at an earlier stage is essential for a better prognosis [2].

Breast cancer is a complex disease with multiple factors contributing to its development. For example, inherited mutations in genes such as BRCA1 and BRCA2 or exposure to toxic substances such as tobacco, smoke, and alcohol can increase a person’s risk of developing breast cancer [3,4,5]. The impact of other environmental factors such as lifestyle, diet, physical activity, hormonal influences, and exposure to radiation on breast cancer development is still being studied [4,5]. It is important to note that the development of breast cancer is influenced by a combination of these factors and further research is needed to fully understand the mechanisms that contribute to this disease.

Carcinogenesis can be initiated by a prolonged history of inflammation, which can be triggered by pathogenic infection, leading to a cascade of inflammatory events and a defective immune response [1,6]. To support this hypothesis, studies show that specific bacterial infections are significant risk factors for carcinomas of different organ systems including the breast, stomach, colon, and pancreas [7,8,9]. For example, a study comparing cancerous and normal ovarian samples showed that a higher percentage of samples in the ovarian cancer group tested positive for *Chlamydia trachomatis* infection in comparison to the control group [10,11]. Additionally, infection with *Helicobacter pylori* plays a crucial role in the development of gastric cancer because of the pathogen’s ability to disrupt the host’s mucosal homeostasis [9]. Further, multiple research groups have found that the advancement of colorectal cancer (CRC) is influenced by intestinal bacteria including H. pylori and the anaerobic bacteria *Fusobacterium nucleatum* [12,13,14,15]. Several studies present that *F. nucleatum* facilitates colorectal carcinoma by impacting the role of tumor-infiltrating lymphocytes and suppressing the action of natural killer (NK) cells, thus producing resistance to chemotherapy [14,15]. It was also revealed that an increased number of *F. nucleatum* in breast cancer cells enhances tumor progression and metastases by impairing the activity of T cells [14]. Another study explored the role of microbiomes in the progression of breast cancer and metastasis [16]. In this study, researchers identified a tumor, called the mouse mammary tumor virus-polyoma middle tumor antigen (MMTV-PyMT), that closely resembles human breast cancer in mice [16]. Researchers observed that breast tumor cells exhibited an oxygen microenvironment that significantly reduced anaerobes while increasing facultative anaerobes [16]. In light of this observation, it was concluded that PyMT spontaneous breast tumors contain significant quantities of live bacteria [16]. In addition, a substantial decrease in lung metastasis was observed following intratumor bacterial reduction by antibiotic treatment administered to eliminate tumor-resident microbiota [16]. These observations suggested that tumor-associated bacteria could assist in cancer spread and metastasis [16]. Additionally, it was shown that microorganisms are capable of inducing DNA damage in breast cancer tissues but not in normal breasts [17].

To further strengthen the above observation, findings from a recent microbiome study showed that several pathogenic bacteria were present in different types of breast cancer tissues [1]. One such microorganism was *Borrelia burgdorferi*, a spirochete that causes Lyme disease. The presence of *B. burgdorferi* in breast cancer tissues was also found to be associated with poor prognosis [1]. This information explains the need to further investigate the influential role of *B. burgdorferi* in the development of breast cancer.

Lyme disease is a vector-borne human disease that affects various body systems, causing musculoskeletal, neurological, and cardiac conditions [18]. *B. burgdorferi* manipulates the mammalian host cells after its entry by modifying its glycoproteins, which help it to invade and propagate in different body tissues [18,19]. It was reported that *B. burgdorferi* could invade human brain cells in vitro and induce an inflammatory process [19]. Further, it was also observed that by the mechanism of intracellular localization *B. burgdorferi* could stay hidden from the host’s immune system [19]. Our research group has recently found evidence that *B. burgdorferi* can invade the triple-negative MDA-MB-231 breast cancer cells at a significantly higher rate in comparison to normal mammary epithelial cells [20]. This study also showed that the in vitro invasion and migratory capability of breast cancer cells was significantly increased following *B. burgdorferi* infection [20]. Our current research focused on identifying the molecular changes that occur when *B. burgdorferi* infects breast tissues, and whether these changes could initiate or progress tumor development. In recent years, microRNAs (miRNA or miR) have been identified as novel cancer biomarkers that can play a vital role in the regulation of molecular mechanisms leading to tumorigenesis [21]. Therefore, the goal of this study was to identify changes in miRNA expression levels in *B. burgdorferi*-infected versus uninfected breast cancer cells to better understand the potential relationship between this infection and breast cancer development.

miRNAs are small non-coding RNAs that post-transcriptionally regulate gene expression [21]. miRNAs are involved in regulating many molecular pathways, and their dysregulation can lead to disease development, including various cancers [21]. Some miRNAs act as tumor suppressors and exhibit decreased expression in certain cancers, while other miRNAs are considered oncogenes and are expressed at higher levels in some cancer types [22]. miRNAs are also implicated in cancer-related processes such as inflammation and tumor progression [22,23]. Numerous studies have reported that the development of breast cancer involves the dysregulated expression of miRNAs [21]. Furthermore, previous research has reported that *B. burgdorferi* infection can affect the expression of miRNAs in host cells [24,25]. It is thus essential to understand the role of miRNAs in *B. burgdorferi*-infected breast carcinoma pathogenesis.

This study focused on analyzing the expression levels of several cancer-associated miRNAs in breast cancer cells following *B. burgdorferi* infection. These potential target miRNAs were selected based on results from an exploratory miRNA breast cancer array and a scientific literature review. A confirmatory study was carried out using quantitative real-time reverse transcription PCR (qRT-PCR), which is considered a standard method for gene expression measurement. This study also focused on analyzing the pathways regulated by our target miRNAs and their associated genes using DIANA software [26]. This research helps in better understanding the miRNAs that contribute to *B. burgdorferi* infection-induced responses and breast cancer development, as well as their associated pathways and genes.

## 2. Materials and Methods

### 2.1. Breast Cancer Array

A Qiagen human breast cancer miRCURY LNA miRNA Focus PCR panel (Qiagen, Hilden, Germany, Cat No: 339325) was utilized to identify target miRNAs that exhibit differential expression in response to *B. burgdorferi* infection along with a scientific literature review. The breast cancer miRNA panel is a ready-to-use 96-well PCR plate that can be used to quantitate the expression levels of 86 miRNAs specifically associated with breast cancer. This breast cancer array was analyzed by qRT-PCR following the manufacturer’s instructions. The cDNA samples used for this experiment were prepared using RNA extracted from uninfected and *B. burgdorferi*-infected MDA-MB-231 cells. The delta-delta Ct method was used to quantify any changes in miRNA expression, with SNORD49 acting as a reference gene.

### 2.2. Mammalian Cell Culture

The normal mammary epithelial cell line MCF-10A (American Type Culture Collection, ATCC CRL-10317, Manassas, VA, USA) and the triple-negative breast cancer cell line MDA-MB-231 (ATCC HTB-26) were cultured using standard tissue culture conditions as previously reported (Gaur 2021). An additional triple-negative breast cancer cell line, HCC1806 (ATCC CRL-2335), was cultured and maintained in RPMI-1640 media (Sigma Aldrich, Louis, MO, USA) with 10% fetal bovine serum (FBS, Gemini Bio, West Sacramento, CA, USA) supplemented with 1% Penicillin–Streptomycin–Glutamine (PSG, ThermoFisher Scientific, Louis, MO, USA). Before the initiation of infection, cells were synchronized to the G0 phase of the cell cycle by incubating in serum-free media for 10 hours. The cells at G0 phase represent a quiescent phase in which cells are not actively dividing, thus allowing analysis of the initial events of the cell cycle [27]. In addition, synchronizing cells reduces the variability between cells, creating a more precise way of comparing experimental results and achieving increased reproducibility of experiments [27]. Following synchronization, a total of one million cells were infected with *B. burgdorferi* with a multiplicity of infection (MOI) of 60 as previously described [20].

### 2.3. Bacterial Culture

Low passage isolates (<6) of *B. burgdorferi* B31 strain (ATCC #35210) were utilized for infection following our previously published protocol (Gaur 2021). All experiments were conducted in three biological replicates (N = 3); four uninfected and four *B. burgdorferi*-infected cell samples from independent rounds of infection were collected and analyzed for each breast cell type studied (N = 12).

### 2.4. Total RNA Extraction

Total RNA from MCF-10A, MDA-MB-231, and HCC1806 cell samples (1 × 10^6^ cells/sample) was extracted using a Qiagen miRNeasy Mini Kit (Qiagen Cat: 217004) following the manufacturer’s instructions. The extracted RNA was stored at −80 °C until further experimental application.

### 2.5. cDNA Synthesis/Quantitation

Single-stranded cDNA was synthesized using a miRCURY LNA RT Kit (Qiagen, 339340) following the manufacturer’s instructions. Each RT reaction required 200 ng of input RNA. cDNA was then quantitated using the Qubit 2.0 Fluorometer and Qubit ssDNA Assay Kit as recommended by the kit protocol (ThermoFisher Scientific, Q10212).

### 2.6. qPCR Analysis of miRNA

qPCR was used to analyze changes in the expression of target miRNAs between uninfected and infected breast cell samples. A miRCURY LNA SYBR Green PCR kit (Qiagen, 339345) and miRNA-specific primers (miRCURY LNA miRNA PCR Assays, Qiagen, 339306) were used to perform qPCR as per the manufacturer’s instructions. Each qPCR reaction was prepared using 60 ng of template cDNA. The expression of target miRNAs was measured in quadruplets for each of the three biological replicates (N = 12).

### 2.7. Calculating miRNA Expression Fold Changes and Statistical Analysis

miRNA expression fold changes were calculated by first obtaining the Cq (cycle of quantitation) values from each qPCR reaction. A total of 12 Cq values were obtained for each target miRNA, from which the three outliers based on the highest and lowest values were excluded from data analysis. Delta-delta Ct values and the expression fold change of each target miRNA between the uninfected and infected samples were calculated using the Microsoft Excel program (Microsoft, Redmont, WA, USA). Statistical analysis for the calculated expression fold changes was performed using a paired, two-tailed Student’s *t*-test in the Qiagen miRCURY LNA miRNA PCR data analysis tool (Qiagen). The data were normalized using the normalization method geNorm (pre-defined reference miRNA only). Using this method, data related to miRNA expression can be normalized based on reference miRNAs in a more accurate and reliable way. A stability measure of <1.5 was returned for the reference gene SNORD48, which was found to be the steadiest reference gene. The expression fold change of a miRNA with a *p*-value of <0.05 was considered statistically significant.

### 2.8. DIANA Analysis of Target miRNAs

The mirPath v.3 DIANA software and DIANA-TarBase v7.0 were used to identify cellular pathways and genes associated with the target miRNAs (https://dianalab.e-ce.uth.gr/html/mirpathv3/index.php?r=mirpath (accessed on 3 January 2023). 

## 3. Results

The goal of this study was to identify any expression fold changes in cancer-associated miRNA molecules between *B. burgdorferi*-infected and uninfected triple-negative breast cancer cell lines MDA-MB-231 and HCC1806. miRNA expression level changes in response to infection were further compared between the two triple-negative breast cancer cell lines and the normal mammary epithelial cell line MCF10A. Along with a published literature review, we performed an exploratory study using a cancer-specific miRNA PCR panel to identify miRNAs with the greatest amount of differential expression in *B. burgdorferi*-infected breast cancer cells compared to the uninfected control [28,29,30,31]. These data led to the identification of four potential microRNAs for evaluation in our study using real-time PCR and pathway analysis.

### 3.1. Breast Cancer Array

In an early exploratory experiment, we used a qPCR array from Qiagen to identify miRNAs that are differentially expressed in *B. burgdorferi*-infected MDA-MB-231 cells compared to uninfected control cells (miRCURY LNA miRNA Focus PCR Panels www.qiagen.com (accessed on 10 December 2022). Results obtained from this array showed that the expression of 38 miRNAs was increased, while the expression of 48 miRNAs was decreased in *B. burgdorferi*-infected MDA-MB-231 cells (Appendix A). Further analyses showed that seven miRNAs were upregulated by >1.5-fold (Table 1), and ten miRNAs were downregulated by >1.5-fold (Table 2).

### 3.2. Selection of Target miRNAs

Target miRNAs were selected based on results from the exploratory Qiagen breast cancer array panel and a review of published scientific literature. The breast cancer array data were used to identify four target miRNAs (miR-206, 214-3p, 16-5p, and 20b-5p) that exhibited the most differentiated Cq values between uninfected and *B. burgdorferi*-infected cells. All the miRNAs selected were found to be related to either breast cancer or reported to be dysregulated in association with *B. burgdorferi* infection (Table 3). To confirm that these four miRNAs were indeed dysregulated in *B. burgdorferi*-infected breast cancer cells compared to the uninfected control, our target miRNAs were further investigated by quantitative real-time PCR.

### 3.3. Selection of a Reference Gene

A reliable reference gene that shows stable expression under both experimental and control conditions is necessary for data normalization and obtaining reliable expression fold change values [38,39]. Our study used a small nucleolar RNA, SNORD48, as a reference gene because of its undifferentiated expression and stability throughout the qRT-PCR experiments performed. Moreover, our scientific literature review identified SNORD48 as an ideal reference gene for analyzing the expression of miRNAs by qRT-PCR [39,40].

### 3.4. Validation of the Differential Expression of Four Selected miRNAs in B. burgdorferi-Infected Breast Cell Lines by qRT-PCR

Differential expressions of the four target miRNAs identified above were validated by comparing the expression levels of each miRNA between *B. burgdorferi*-infected and uninfected breast cells. Expression levels of each target miRNA were measured in uninfected and infected breast cancer cell lines HCC1806, MDA-MB-231, and normal breast cell line MCF10A using qRT-PCR. Differences in miRNA expression levels between the uninfected and infected samples are outlined in Table 4, Table 5 and Table 6 below.

The largest difference in miRNA expression identified between uninfected and infected samples was observed for miR-206, which was significantly upregulated by approximately 5-fold in both triple-negative breast cancer cell lines (Table 4 and Table 5, Figure 1 and Figure 2). Interestingly, miR-206 was also slightly but significantly upregulated in the normal breast cell line MCF10A by 0.6-fold (Table 6, Figure 3). A very similar trend was found for miR-214-3p which was significantly upregulated in all three cell lines, such as miR-214-3p which was observed to be upregulated by approximately 2-fold in the MDA-MB-231 and HCC1806 cell samples, and 0.8-fold in the MCF10A cell samples (Table 4, Table 5 and Table 6, Figure 1, Figure 2 and Figure 3). A small but statistically significant change in the expression of miR-16-5p was observed only in the MDA-MB-231 cell line. A minor, statistically significant change in miR-20-5p expression was also found in all three cell lines, in the range of 0.8–1.3-fold (Table 4, Table 5 and Table 6, Figure 1, Figure 2 and Figure 3).

### 3.5. DIANA Analysis of Target miRNAs

The mirPath v.3 DIANA software was used to investigate the cellular pathways associated with each of the target miRNAs chosen in this study. Further assessment was performed using KEGG analysis and the Tarbase v7.0 database.

Multiple statistically significant pathways (*p*-value of <0.05) associated with each target miRNA were identified. Table 7 summarizes the pathways associated with each target miRNA and their corresponding *p*-values. The observed data show that the Gap junction pathway is most significantly affected by miR-206, followed by the cell cycle, mismatch repair, and the ECM (extracellular matrix) receptor interaction pathways. miR-214-3p most significantly affects the ECM–receptor interaction pathway but is also associated with the GAP junction, microRNAs in cancer, and additional cancer pathways. The cell cycle, p53 signaling pathway, and pathways in cancer were observed to have a significant association with miR-16-5p. For miR-20b-5p, the p53 signaling pathway and pathways in cancer were found to be statistically significant.

Next, shared pathways were identified using DIANA which was further analyzed to identify those associated with more than one of the four target miRNAs (Table 8). The GAP junction and ECM–receptor interaction pathways were found to be affected by both miR-206 and 214-3p. The cell cycle pathway has been associated with miR-206 and 16-5p. Finally, the p53 signaling pathway was observed to be affected by miR-16-5p and 20b-5p. Pathways in cancer were alternatively associated with three out of four target miRNAs, miR-214-3p, 16-5p, and 20b-5p. In summary, pathways in cancer are the only pathway affected by more than two of the four target miRNAs. The remaining two pathways, mismatch repair and microRNAs in cancer, are each influenced by only one target miRNA: miR-206 and 214-3p, respectively.

The pathways associated with each target miRNA were further analyzed to identify the specific genes affected by miRNA interaction; these genes are outlined in Figure 4. The ECM–receptor interaction pathway gene Fibronectin-1 protein (FN1) has been found to be affected by both miR-206 and 214. miR-206 and 214 also affect the Gap junction pathway genes Gap Junction Protein Alpha 1 (GJA1) and Inositol 1,4,5-trisphosphate receptor type 3 (ITPR3). miR-214 and 206 are each associated with additional genes related to other pathways. Two pathways associated with miR-214-3p, microRNAs in cancer and pathways in cancer, were both observed to involve the gene Phosphatase and Tensin homolog (PTEN). miRNA-214-3p is also associated with Mitogen-Activated Protein Kinase1 (MAPK1) in relation to the microRNAs in the cancer pathway. miR-206 has been associated with the Exonuclease 1 (EXO1) gene, which is related to the mismatch repair pathway. The cell cycle pathway-related genes WEE1 G2 Checkpoint Kinase (WEE1) and Protein Kinase Membrane Associated Tyrosine/Threonine 1 (PKMYT1) were both associated with miR-206. Different cell cycle-related genes including MDM2 (Murine Double Minute 2) and CDC25A (Cell Division Cycle 25A) were alternatively found to be affected by miR-16-5p. miR-16-5p and 20b-5p were found to be associated with STAT3 (Signal Transducer and Activator of Transcription 3), a gene related to pathways in cancer.

## 4. Discussion

Previous research has demonstrated that the dysregulation of miRNA expression plays a significant role in breast cancer development [21,28,29,30,31]. Other reports have indicated that *B. burgdorferi* infection can affect the expression of certain miRNAs in host cells [25]. Therefore, we aimed to understand the impact of *B. burgdorferi* infection on the expression and function of miRNAs in breast cancer. In this study, miRNAs of interest were first identified from an exploratory breast cancer array and extensive literature reviews [21,28,29,30,31]. The expression of the target miRNAs was then further validated by qRT-PCR.

We found a very significant expression change of two miRNAs, miR-206 and 214-3p, in *B. burgdorferi*-infected triple-negative breast cancer cell lines. Studies have previously demonstrated an upregulated expression of these miRNAs in breast cancer cells [28,41], especially miR-206 which was found to be associated with poor prognosis in breast cancer patients [28]. Another study found that miR-206 facilitates the invasion, migration, and proliferation of breast cancer cells by suppressing the expression of the G-protein-coupled receptor neurokinin-1 (NK1R-FL) [41]. Moreover, the proliferation of breast cancer cells was significantly reduced with the downregulation of miR-206 [41]. These previous studies indicate that miR-206 likely plays a role in inducing a tumorigenic response.

Another important effect of miR-206 is the regulation of matrix metalloproteinase 9 (MMP9) expression, a protein that acts in tissue reassembly by activating cytokines and chemokines that degenerate the extracellular matrix [32]. In a mycobacterial infection study, miR-206 was found to regulate the expression of MMP9 by influencing tissue inhibitors of metalloproteinases such as TIMP3 [33]. Another study identified an upregulated expression of miR-206 in the failing hearts of mice and cardiac fibroblast cells and was found to be associated with reduced TIMP 3 activity [42]. Previously, our research group detected a significant increase in the expression of MMP9 in *B. burgdorferi*-infected MDA-MB-231 cells compared to the uninfected control group [20]. In contrast, the normal mammary epithelial cell line, MCF10A, did not show any change in MMP9 expression after infection with *B. burgdorferi* [20]. This finding demonstrated that the migratory and invasive characteristics of breast cancer cells were influenced by the level of tissue remodeling markers following infection with *B. burgdorferi* [20].

miR-214-3p also exhibited significant upregulation in our study which is supported by previous studies that have found miR-214-3p upregulation to be associated with the progression of triple-negative breast cancer [29,43]. One study observed that miR-214 promotes breast cancer progression by impairing the phosphoinositide 3-kinase (PI3K)/protein kinase B (Akt)/mammalian target of rapamycin (mTOR) signal transduction (PI3K/Akt/mTOR) pathway [29]. This pathway is one of the most important cell signaling pathways as it plays an essential role in regulating cell proliferation, apoptosis, and differentiation [29]. Another study indicated that miR-214 was significantly upregulated in sporadic breast cancer patients compared to the hereditary form [43]. Furthermore, triple-negative breast cancer patients with a higher expression of miR-214 exhibited a poorer prognosis than patients with a lower expression of miR-214 [43]. In yet another study, miR-214 was among the miRNAs found to be differentially expressed in primary astrocytes after infection with *B. burgdorferi* [35]. Moreover, a research study observed that triple-negative breast cancer cells’ viability, migration, and invasion capacity were higher with miR-214-3p upregulation and that inhibiting this miRNA suppressed these effects [44]. Furthermore, miR-214-3p was found to target ST6GAL1 by negative feedback regulation [44]. ST6GAL1 is a sialyltransferase whose transformation relates to the growth, invasion, and metastasis of the tumor [44,45]. It was identified that knocking down ST6GAL1 repressed triple-negative breast cancer cells’ viability, migration, and invasion abilities, which were restored by lowering the expression of miR-214-3p, indicating the association of this miRNA in breast cancer development [44].

In our study, the remaining targets, miR-16-5p and 20b-5p, did not demonstrate drastic expression changes compared to miR-206 and 214-3p, but were still statistically significant in the evaluated breast cell lines. A recent study published in 2021 concluded that the overexpression of miR-16-5p in breast cancer tissues resulted in cell cycle arrest and inhibited breast cancer cell proliferation [30]. An upregulation of ANLN (Anillin actin-binding protein) was identified in breast carcinoma and was a target of miR-16-5p in breast cancer cells [30]. ANLN is involved in cell division and is upregulated in various cancers such as pancreatic and cervical cancer [46,47]. Furthermore, it was discovered that miR-16-5p suppressed ANLN expression contributing to the inhibition of breast cancer cells, similar to the overexpression of miR-16-5p [30]. Additionally, it was reported that infection with *Helicobacter pylori* (*H. pylori*) induces an inflammatory process by the persistent colonization of the gastric mucosa, which could ultimately progress to more severe diseases such as gastric carcinoma [36,48]. Research also demonstrated that infection of the gastric epithelial cells resulted in an upregulation of miR-16, suggesting the role of bacterial infection in the dysregulation of this miRNA [48]. miR-20b-5p, evaluated in our study, is also implicated in playing a critical role in the development of breast cancer [31,37]. One study observed that miR-20b-5p significantly promoted breast cancer cell migration and proliferation capacity [31]. Researchers also demonstrated that upregulation of miR-20b-5p in breast cancer stem cells (BCSCs) inhibited their apoptosis and facilitated proliferation [31]. Further, an in vivo experiment involving xenograft tumors of breast cancer cells indicated that miR-20b-5p overexpression led to their significant enlargement [31]. Another study revealed the overexpression of miR-20b in breast cancer cells inhibited a critical tumor suppressor gene, PTEN, by binding to its 3′-UTR [37]. Additionally, the upregulation of miR-20b significantly promoted the proliferation and colony formation of breast cancer cells, whereas its downregulation reduced the growth of breast cancer cells [37]. PTEN downregulation activates the critical oncogenic PI3K/Akt signaling pathway, hence regulating cell proliferation, migration, cell cycle, and apoptosis during the development of cancer [37,49].

Our study further analyzed the pathways and genes associated with each target miRNA using the mirPath v.3 DIANA software. Multiple pathways were found to be associated with each miRNA studied, and each of these pathways involved multiple genes regulated by the corresponding miRNA. For example, miR-206 and 214-5p were found to influence two pathways, ECM–receptor interaction and Gap junction. The ECM–receptor interaction pathway involves communication between cells and the extracellular matrix, a process that assists in the development of cell–matrix interactions and ECM remodeling [50]. Mutation in this pathway can result in the development of a tumor, augment the migration of tumor cells, and initiate metastatic progression by redesigning the ECM in distant organs [50]. Relevantly, one research study has demonstrated a direct association between ECM remodeling and breast cancer progression [51]. The study identified advanced tumor growth caused by increased collagen-I deposition in mouse mammary epithelial cells, along with aggressive tumor progression and invasive lung metastasis [51]. In both miR-206 and 214-5p, the genes FN1 and GJA1 were found to be associated with the ECM–receptor interaction and Gap junction pathway, respectively [26].

The Gap junction pathway, regulated by miR-206 and 214-5p, consists of intercellular channels that lead to the direct transmission of ions and small molecules between the cytosolic compartments of adjacent cells [52]. The upregulated expression of the FN1 gene has been correlated with the development of breast cancer and poor prognosis [53]. It has also been demonstrated that, when upregulated, miR-206 targets and leads to the irregular expression of the FN1 gene in Hirschsprung disease patients [54]. Moreover, *B. burgdorferi* targets the FN1 protein by expressing BBK32, a fibronectin-binding protein [55]. An upregulated expression of the GJA1 gene has been demonstrated to play a significant role in the vesicle-mediated transport of the transcellular location of *Borrelia bavariensis* and resulting in infection of the human brain microvascular endothelial cells [56]. This report, therefore, contributes to a better understanding of the resulting molecular changes caused by the dysregulation of miR-206 and 214-5p related to pathways and the associated genes [53,55,56]. Further, our data support the hypothesis that changes in miRNA expression can be a contributing factor to the development of breast cancer after *B. burgdorferi* infection although more research should investigate this connection further.

In addition, miR-206 is associated with two other pathways, cell cycle, and mismatch repair, suggesting it may play a role in maintaining cellular homeostasis [26]. According to the DIANA analysis, the cell cycle is influenced by genes WEE1 and PKMYT1 which play a vital role in recognizing and repairing DNA damage [26,57]. Both these genes exhibit an upregulated expression in breast cancer cells and act as oncogenes, assisting in further growth and carcinoma progression [57]. It has been reported that as malignant transformation is induced, WEE1 upregulation in cancer cells might promote tumorigenesis by maintaining the levels of genomic instability [57]. The DIANA analysis further shows that the mismatch repair pathway, regulated by miR-206, involves the EXO1 gene, which regulates the mechanisms of DNA damage checkpoints and repair [58]. The EXO1 gene was investigated in a study using MDA-MB-231 breast cancer cells where an increase in gene expression was observed compared to the normal mammary epithelial cell line, MCF10A [58]. This report suggests that the elevated expression of this EXO1 gene is associated with breast cancer development and poor prognosis by affecting the mismatch repair pathway of the cells [58]. The role of miR-206 may be a contributing factor in carcinogenesis, impaired cellular function, and DNA damage through its relation to the cell cycle and mismatch repair pathway [26].

DIANA analysis of miR-214-3p shows that this miRNA can influence two other pathways, microRNAs in cancer and pathways in cancer. Both these pathways involve the PTEN and MAPK1 genes [26]. PTEN mostly acts as a tumor suppressor, and it has been observed that loss of PTEN expression is associated with breast cancer progression [59]. Further, a research study has shown that in breast cancer cells, the upregulated expression of the MAPK pathway is associated with reduced PTEN expression [59]. Through DIANA analysis, miR-214-3p appears to influence another gene, the MDM2, which is involved in cancer pathways [26]. There is evidence that the members of the MDM family play a crucial role in regulating the fate of cell development [60]. This function of MDM is achieved by the negative control of an essential tumor suppressor gene, p53 [60]. In addition, research has concluded that the overexpression of the MDM2 gene is related to the augmentation of breast cancer [60]. Hence, the role of miR-214-3p in these pathways obtained by DIANA analysis describes its potential effect on breast cancer development [26]. In this study, a higher expression of miR-206 and 214-3p in the *B. burgdorferi*-infected triple-negative breast cancer cells was confirmed which provided strong evidence that these miRNAs may play a crucial role in the spirochetal infection of mammalian epithelial cells.

The DIANA analysis for miR-16-5p showed that it affects three pathways, one of which is the cell cycle pathway associated with the genes MDM2 and CDC25A [26]. The CDC25A gene regulates the G1/S and G2/M checkpoints of the cell cycle, and its dysregulation has been implemented in the development of triple-negative breast cancer [61]. Thus, miR-16-5p might be regulating genes involved in cell cycle pathways. Further DIANA analysis identified that miR-16-5p and 20b-5p share two other pathways. Both miRNAs were found to be associated with pathways in cancer and the p53 signaling pathway coded by the STAT3 and MDM2 genes [26]. STAT3 regulates the progress of breast cancer and the proliferation of cells by targeting oncogenes [62]. As mentioned earlier, miR-16-5p and 20b-5p were associated with the p53 signaling pathway, which involves the p53 tumor suppressor gene, whose dysregulation led to the growth of more aggressive breast cancer and poor prognosis [63].

In our DIANA analysis, it was further seen that out of the four target miRNAs, the MDM2 gene was associated with three miRNAs, miR-214-3p, 16-5p, and 20b-5p [26]. MDM2 is found to act as an oncoprotein mainly by degrading the tumor suppressor gene p53 via E3 ubiquitin ligase action [60,64]. The association of our target miRNAs with MDM2 has been a significant observation as this gene is found to be dysregulated in several cancers [60,64]. In addition, the pathway in cancer, as seen in miR-16-5p and 20b-5p, was also observed in miR-214-3p, making it the only pathway that is observed to be influenced by three miRNAs [26]. Hence, by analyzing the miRNA pathways using the DIANA, it was established that most of these target miRNAs influence cellular mechanism pathways associated with oncogenes, suggesting a possible role in the development and progression of breast cancer cells and *B. burgdorferi* infection.

## 5. Conclusions

In summary, this research study demonstrated that *B. burgdorferi* infection can cause the dysregulation of several miRNAs as observed in normal and tumorigenic breast cancer cell lines. The most statistically significant upregulations were observed for miR-206 and 214-3p in *B. burgdorferi*-infected breast cell lines compared to the uninfected breast cells. However, miRNAs can target hundreds of genes, thus, even the slight elevation of their expression can severely impact gene regulation. miRNA pathway analysis using DIANA also emphasized their potential role in regulating multiple oncogenes and augmenting the development of breast cancer. In addition, these miRNAs could be eventually targeted as a tool for therapeutic and diagnostic purposes for the analysis of breast cancer progression and management. Consequently, the miRNA molecules and their pathways with associated genes should be further evaluated to understand their role in breast cancer and the influence of *B. burgdorferi* infection in a more extensive form.

## Figures and Tables

**Figure 1 microorganisms-11-01475-f001:**
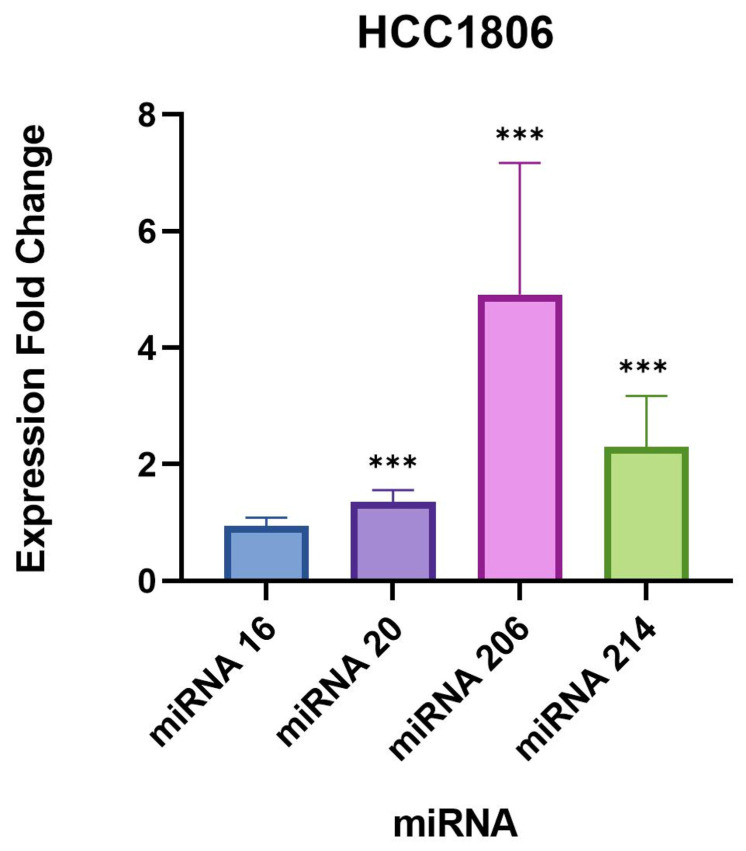
Expression fold changes of the target miRNAs in B. burgdorferi-infected HCC1806 triple-negative breast cancer cells compared to the uninfected HCC1806 cells. ***, *p*-value < 0.05.

**Figure 2 microorganisms-11-01475-f002:**
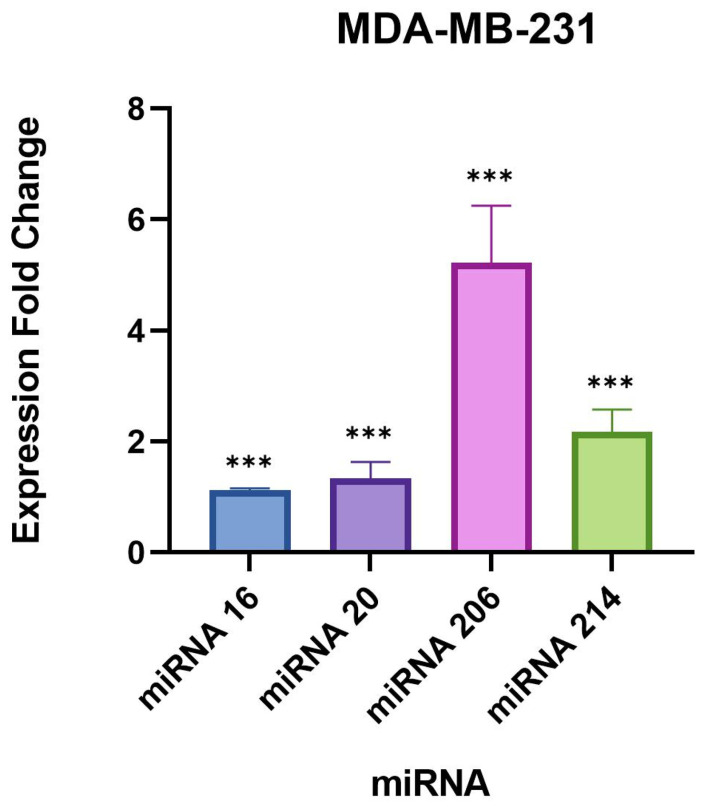
Expression fold changes of the target miRNAs in *B. burgdorferi*-infected MDA-MB-231 triple-negative breast cancer cells compared to the uninfected MDA-MB-231 cells. ***, *p*-value < 0.05.

**Figure 3 microorganisms-11-01475-f003:**
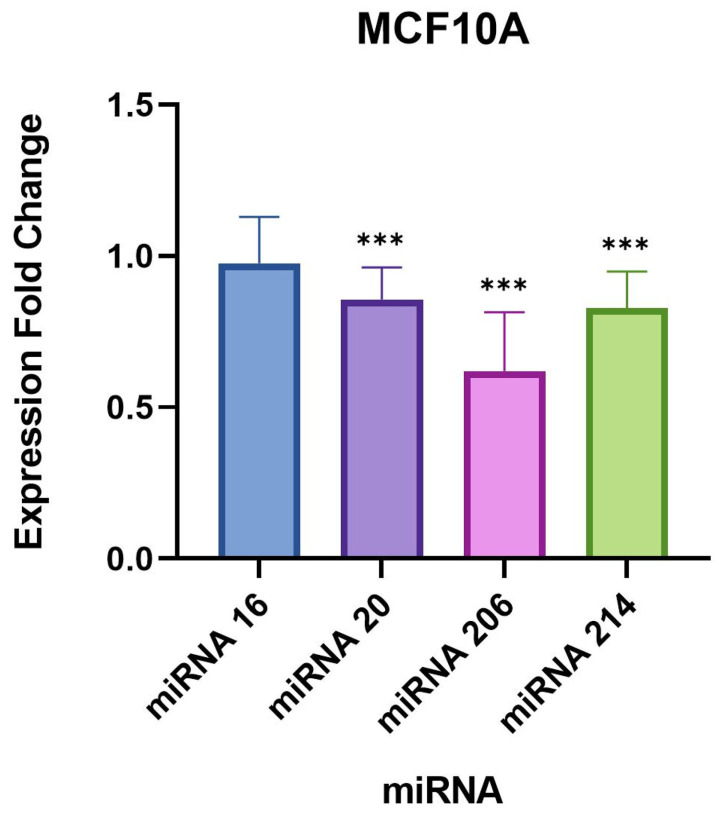
Expression fold changes of the target miRNAs in *B. burgdorferi*-infected MCF10A non-tumorigenic mammary epithelial cells compared to the uninfected MCF10A cells. ***, *p*-value < 0.05.

**Figure 4 microorganisms-11-01475-f004:**
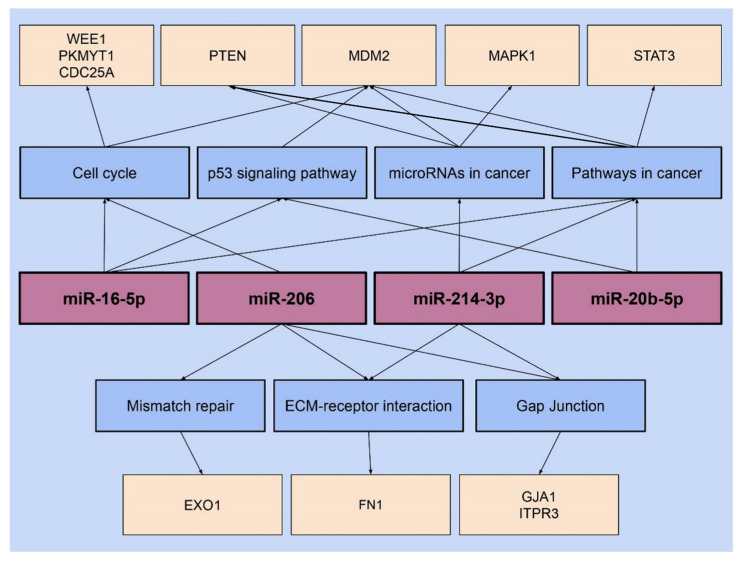
miRNA pathways and the related genes of each target miRNA through DIANA analysis.

**Table 1 microorganisms-11-01475-t001:** List of seven miRNAs observed to be upregulated in MDA-MB-231 breast cancer cells after infection with *B. burgdorferi* based on data from a Qiagen breast cancer microRNA panel.

Upregulated miRNAs	
miRNA	Fold Change
miR-206	26.06
miR-202-3p	3.92
miR-214-3p	2.66
miR-20b-5p	2.11
miR-146a-5p	1.73
miR-16-5p	1.6
miR-101-3p	1.56

**Table 2 microorganisms-11-01475-t002:** List of 10 miRNAs observed to be downregulated in MDA-MB-231 breast cancer cells after infection with *B. burgdorferi* based on data from a Qiagen breast cancer microRNA panel.

Downregulated miRNAs	
miRNA	Fold Change
miR-22-3p	−1.55
miR-423-5p	−1.55
miR-133a-3p	−1.56
miR-29b-3p	−1.56
let-7b-5p	−1.63
miR-155-5p	−1.66
miR-200a-3p	−1.99
miR-181a-5p	−2.14
miR-145-5p	−2.16
miR-143-3p	−2.4

**Table 3 microorganisms-11-01475-t003:** List of target miRNAs with a focus on their associations with breast cancer and bacterial infection.

miRNA	General Overview	Role in Breast Cancer/Infections
miR-206	Oncogene	Upregulated expression of this miRNA has been found in breast cancer [28]. Associated with MMP9 and acts in the hemorrhagic transformation of cardioembolic stroke [32]; and also plays a role in TIMP3 induction, with mycobacterial infection observed [33]. Observed upregulated expression with mycobacterial infection [34].
miR-214-3p	Oncogene	Upregulated expression associated with the progression of triple-negative breast cancer [29]. Affects the PI3K/Akt/mTOR pathway and promotes breast cancer development [29]. Differentially expressed in *B. burgdorferi*-infected primary human astrocytes [35].
miR-16-5p	Tumor suppressor	The upregulated expression leads to the inhibition of the progress of the cell cycle and the proliferation of breast cancer cells [30]. Dysregulated expression has been found in *H. Pylori*-infected gastric epithelial cells [36].
miR-20b-5p	Oncogene	Upregulated expression results in increased migration and proliferation of breast cancer cells [31]. Promotes tumor development and influences PTEN gene expression in breast cancer [37].

**Table 4 microorganisms-11-01475-t004:** Expression fold change of target miRNAs in *B. burgdorferi*-infected HCC1806 triple-negative breast cancer cells compared to uninfected control cells. qRT-PCR data were normalized using SNORD48 as a reference gene. Expression fold changes with a *p*-value of <0.05 were considered statistically significant.

miRNA	Fold Change	*p*-Value
hsa-miR-206	4.92	1.34 × 10^−3^
hsa-miR-214-3p	2.22	2.66 × 10^−3^
hsa-miR-16-5p	0.94	3.89 × 10^−1^
hsa-miR-20b-5p	1.34	4.65 × 10^−3^

**Table 5 microorganisms-11-01475-t005:** Expression fold changes of target miRNAs in *B. burgdorferi*-infected MDA-MB-231 triple-negative breast cancer cells compared to uninfected control cells. qRT-PCR data were normalized using SNORD48 as a reference gene. Expression fold changes with a *p*-value of <0.05 were considered statistically significant.

miRNA	Fold Change	*p*-Value
miR-206	5.13	3.00 × 10^−6^
miR-214-3p	2.14	0.00
miR-16-5p	1.12	9.70 × 10^−5^
miR-20b-5p	1.3	2.16 × 10^−2^

**Table 6 microorganisms-11-01475-t006:** Expression fold changes of target miRNAs in *B. burgdorferi*-infected MCF10A normal mammary epithelial cells compared to uninfected control cells. qRT-PCR data were normalized using SNORD48 as a reference gene. Expression fold changes with a *p*-value of <0.05 were considered statistically significant.

miRNA	Fold Change	*p*-Value
miR-206	0.6	8.74 × 10^−4^
miR-214-3p	0.82	6.30 × 10^−3^
miR-16-5p	0.97	6.07 × 10^−1^
miR-20b-5p	0.85	8.44 × 10^−3^

**Table 7 microorganisms-11-01475-t007:** Cellular pathways associated with target miRNAs and their statistical significance (*p*-value < 0.05) obtained from DIANA analysis.

miRNA	Pathway	*p*-Value
miR-206	GAP junction	6.48 × 10^−6^
	Cell cycle	0.000197
	Mismatch repair	0.003945
	ECM–receptor interaction	0.03367
miR-214-3p	ECM–receptor interaction	3.40 × 10^−17^
	GAP junction	7.14 × 10^−7^
	microRNAs in cancer	0.004235964
	Pathways in cancer	0.036322571
miR-16-5p	Cell cycle	3.38 × 10^−6^
	P53 signaling pathway	0.000887863
	Pathways in cancer	0.001992202
miR-20b-5p	P53 signaling pathway	0.0054649
	Pathways in cancer	0.012188139

**Table 8 microorganisms-11-01475-t008:** Representation of common pathways associated with target miRNAs identified using DIANA analysis. ✓: means Present.

Pathway	miR-206	miR-214-3p	miR-16-5p	miR-20b-5p
GAP junction	✓	✓	-	-
ECM–receptor interaction	✓	✓	-	-
Cell cycle	✓	-	✓	-
Pathways in cancer	-	✓	✓	✓
p53 signaling pathway	-	-	✓	✓
Mismatch repair	✓	-	-	-
MicroRNAs in cancer	-	✓	-	-
Total no of pathways	4	4	3	2

## Data Availability

The data presented in this study are available at a reasonable request from the corresponding author.

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
