# Peer review of "Effect of Borrelia burgdorferi on the Expression of miRNAs in Breast Cancer and Normal Mammary Epithelial Cells"

_microorganisms, 2023, doi:10.3390/microorganisms11061475_

Round 1

Reviewer 1 Report

Dear authors, congratulations for the work I have no comments to report.

I believe that the study is well done and written correctly. The authors has been shown that B. burgdorferi can induce changes in miRNA expression in breast cancer cells and this is an important starting point for future studies, especially in areas endemic for Borrelia burgdorferi. Comment: Histogram images are grainy, I suggest the authors to insert sharper.  

Author Response

Dear Reviewer,

Thank you for reviewing our manuscript. We appreciate your time and your positive feedback. In response to your comment about the poor quality of the figures, we have made the necessary changes. 

We hope that our revisions have adequately addressed your concern, and we look forward to receiving your further feedback.

Reviewer 2 Report

In this manuscript, the authors reported that  Borrelia burgdorferi can induce miRNA expression changes in breast cancer cells, which can further effect breast cancer related pathways. This manuscript can be published after minor revision:

1. In Table 1 and 2, both fold-change and p-values should be listed. Plus, for a clearer illustration, please order the miRNAs according to the absolute fold change values.

2. In Section Selection of targeted miRNAs, the author identified four miRNAs according to the results obtained in breast cancer array experiment. The criteria of selection was not clearly demonstrated.

1) The second up-regulated miRNA is miR203-p (fc=3.91). Why was this miR not selected but the third up-regulated miR (miR-214-3p, fc=2.66) was selected?

2) How did the authors determine the number of miRs to be selected?

3) Why were not any downregulated miR selected?

2. The quality of figures should be improved.

In this manuscript, the authors reported that  Borrelia burgdorferi can induce miRNA expression changes in breast cancer cells, which can further effect breast cancer related pathways. This manuscript can be published after minor revision:

1. In Table 1 and 2, both fold-change and p-values should be listed. Plus, for a clearer illustration, please order the miRNAs according to the absolute fold change values.

2. In Section Selection of targeted miRNAs, the author identified four miRNAs according to the results obtained in breast cancer array experiment. The criteria of selection was not clearly demonstrated.

1) The second up-regulated miRNA is miR203-p (fc=3.91). Why was this miR not selected but the third up-regulated miR (miR-214-3p, fc=2.66) was selected?

2) How did the authors determine the number of miRs to be selected?

3) Why were not any downregulated miR selected?

2. The quality of figures should be improved.

Author Response

Dear Reviewer,

Thank you for reviewing our manuscript. We appreciate your time and valuable feedback. In response to your comments and questions, we have made the necessary revisions to address the concerns raised.

  1. Regarding Tables 1 and 2, we apologize for the oversight in ordering the miRNAs according to the absolute fold change values. We have rectified this error and reorganized the miRNAs in both tables based on their fold change values in descending order.

As for the p-values in Tables 1 and 2, we would like to clarify that the data presented in these tables were derived from an exploratory experiment that was conducted as a preliminary analysis to identify potential miRNAs associated with Borrelia burgdorferi-infected breast cancer cells. Due to the nature of this exploratory study, the p-value was not established for this dataset since it was obtained from only one independent experiment (N=1). MiRNA-specific Quantitative Real-Time Reverse Transcription PCR was used to confirm and statistically analyze the obtained data and establish the necessary p-values using sample sizes N=12.

It is important to note that the exploratory study data displayed in Tables 1 and 2 served as supplementary evidence alongside a thorough review of the scientific literature [1-10] in the process of selecting target miRNAs for the final aim of our study.

  1. Regarding the selection of target miRNAs, it was elaborated upon in the Results Section: 3.2, titled "Selection of targeted miRNAs." In this section, we explicitly stated that the target miRNAs were chosen based on the results of our exploratory experiment using the Qiagen breast cancer array panel, in conjunction with a comprehensive review of published scientific literature [1-10].

The breast cancer array helped identify multiple upregulated target miRNAs as an exploratory analysis. Finally, the final four target miRNAs were chosen based on their relationship with breast cancer and bacterial infection, which were comprehended by an extensive scientific literature review [1-10]. To support this claim, we have provided the relevant data from the breast cancer array in Tables 1 and 2. Additionally, in Table 3, we included an overview of the association between the target miRNAs in breast cancer and bacterial infection by referencing established scientific literature [1-10].

  1. The selection of miR-214-3p over miR-203-p (fold change (fc) = 2.66 and 3.91, respectively) was primarily based on a combination of factors related to their differential expression and functional relevance in the context of our study objectives. Although miR-203-p exhibited a higher fold change, its functional significance in breast cancer and Borrelia burgdorferi infection studies was significantly less established. On the other hand, miR-214-3p, despite having a slightly lower fold change, has been extensively reported in the literature [2, 8] to be associated with breast cancer and bacterial pathogen infection. Additionally, miR-214-3p was found to be dysregulated in Borrelia burgdorferi infection, and the multiple molecular pathways implicated in breast cancer studies thus supporting its analysis in relation to our hypothesis. Therefore, considering both the fold change and functional relevance, we deemed miR-214-3p to be a more suitable candidate for further investigation. However, in our future studies, we will fully investigate miR-203-p involvement Borrelia burgdorferi-infected cells.
  2. The determination of the number of miRNAs that were selected for the study was guided by an extensive scientific literature review [1-10] in addition to the explanatory breast cancer array panel result. We selected the four target miRNAs that were found to be most significantly relevant to breast cancer and bacterial infection studies based on the scientific literature review [1-10]. We then confirmed the expression fold change of the final selected four miRNAs by conducting Real-Time PCR. The expression fold change of the final four target miRNAs was measured in quadruplets for each of the three biological replicates (N=12).
  3. The focus of our study was primarily on up-regulated miRNAs due to their potential role in the regulatory mechanisms of breast cancer and bacterial infection. While downregulated miRNAs could certainly contribute to gene regulation in breast cancer and microbiome infection studies, their selection was beyond the scope of our current investigation. Therefore, we decided to narrow down our study to up-regulated miRNAs to maintain a clear research focus and provide comprehensive insights into the upregulation mechanisms underlying breast cancer and Borrelia burgdorferi However, we acknowledge the importance of downregulated miRNAs and their potential implications, and we will address this aspect in future investigations.

                  We sincerely appreciate your insightful feedback, which has significantly contributed to the improvement of our paper. We have carefully addressed each of your concerns and incorporated the necessary changes in the revised version. We believe these modifications have enhanced the overall quality and clarity of our research findings.

                 Once again, we thank you for your valuable input and for your time in reviewing our manuscript. We hope that our revisions have adequately addressed your concerns, and we look forward to receiving your further feedback.

References:

  1. Quan Y, Huang X, Quan X. Expression of miRNA-206 and miRNA-145 in breast cancer and correlation with prognosis. Oncol Lett. 2018;16(5):6638-6642. doi:10.3892/ol.2018.9440
  2. Zhang Y, Zhao Z, Li S, et al. Inhibition of miR 214 attenuates the migration and invasion of triple-negative Breast Cancer cells. Mol Med Rep. 2019;19(5):4035-4042. doi:10.3892/mmr.2019.10112
  3. Wang Z, Hu S, Li X, et al. MiR-16-5p suppresses breast cancer proliferation by targeting ANLN. BMC Cancer. 2021;21(1):1188. Published 2021 Nov 7. doi:10.1186/s12885-021-08914-1
  4. Xia L, Li F, Qiu J, et al. Oncogenic miR-20b-5p contributes to malignant behaviors of breast cancer stem cells by bidirectionally regulating CCND1 and E2F1. BMC Cancer. 2020;20(1):949. Published 2020 Oct 2. doi:10.1186/s12885-020-07395-y
  5. Zheng L, Xiong Y, Liu J, et al. MMP-9-Related microRNAs as Prognostic Markers for Hemorrhagic Transformation in Cardioembolic Stroke Patients. Front Neurol. 2019;10:945. Published 2019 Sep 6. doi:10.3389/fneur.2019.00945
  6. Fu X, Zeng L, Liu Z, Ke X, Lei L, Li G. MicroRNA-206 regulates the secretion of inflammatory cytokines and MMP9 expression by targeting TIMP3 in Mycobacterium tuberculosis-infected THP-1 human macrophages. Biochem Biophys Res Commun. 2016;477(2):167-173. doi:10.1016/j.bbrc.2016.06.038
  7. Wright K, de Silva K, Plain KM, et al. Mycobacterial infection-induced miR-206 inhibits protective neutrophil recruitment via the CXCL12/CXCR4 signaling Chang axis. PLoS Pathog. 2021;17(4):e1009186. Published 2021 Apr 7. doi:10.1371/journal.ppat.1009186
  8. Casselli T, Qureshi H, Peterson E, et al. MicroRNA and mRNA Transcriptome Profiling in Primary Human Astrocytes Infected with Borrelia burgdorferi. PLoS One. 2017;12(1):e0170961. Published 2017 Jan 30. doi:10.1371/journal.pone.0170961
  9. Cui L, Markou A, Stratton CW, Lianidou E. Diagnosis and Assessment of Microbial Infections with Host and Microbial MicroRNA Profiles. Advanced Techniques in Diagnostic Microbiology. 2018;563-597. Published 2018 Nov 10. doi:10.1007/978-3-319-95111-9_23
  10. Zhou W, Shi G, Zhang Q, Wu Q, Li B, Zhang Z. MicroRNA-20b promotes cell growth of breast cancer cells partly via targeting phosphatase and tensin homologue (PTEN). Cell Biosci. 2014;4(1):62. Published 2014 Oct 14. doi:10.1186/2045-3701-4-62